# Rapid and Visual Detection of SARS-CoV-2 RNA Based on Reverse Transcription-Recombinase Polymerase Amplification with Closed Vertical Flow Visualization Strip Assay

Yumeng Song,[a] Pei Huang,[a] Mengtao Yu,[a] Yuanguo Li,[b] Hongli Jin,[a] Jiazhang Qiu,[a] Yuanyuan Li,[a] Yuwei Gao,[b] Haili Zhang,[a] Hualei Wang[a]

[a]Key Laboratory of Zoonosis Research, Ministry of Education, College of Veterinary Medicine, Jilin University, Changchun, China
[b]Changchun Veterinary Research Institute, Chinese Academy of Agricultural Sciences, Changchun

Yumeng Song and Pei Huang contributed equally to this work. Author order was determined by drawing straws.

**ABSTRACT** Coronavirus disease 2019 (COVID-19), which is caused by severe acute respiratory syndrome coronavirus 2 (SARS-CoV-2), was initially identified in 2019, after which it spread rapidly throughout the world. With the progression of the epidemic, new variants of SARS-CoV-2 with faster transmission speeds and higher infectivity have constantly emerged. The proportions of people asymptomatically infected or reinfected after vaccination have increased correspondingly, making the prevention and control of COVID-19 extremely difficult. There is therefore an urgent need for rapid, convenient, and inexpensive detection methods. In this paper, we established a nucleic acid visualization assay targeting the SARS-CoV-2 nucleoprotein (*N*) gene by combining reverse transcription-recombinase polymerase amplification with closed vertical flow visualization strip (RT-RPA-VF). This method had high sensitivity, comparable to that of reverse transcription-quantitative PCR (RT-qPCR), and the concordance between RT-RPA-VF and RT-qPCR methods was 100%. This detection method is highly specific and is not compatible with bat coronavirus HKU4, human coronaviruses 229E, OC43, and HKU1-CoV, Middle East respiratory syndrome coronavirus (MERS-CoV), or other respiratory pathogens. However, multiple SARS-CoV-2 variants are detectable within 25 min at 42°C using this visual method, including RNA transcripts of the Wuhan-Hu-1 strain at levels as low as 1 copy/$\mu$L, the Delta strain at 1 copy/$\mu$L, and the Omicron strain at 0.77 copies/$\mu$L. The RT-RPA-VF method is a simple operation for the rapid diagnosis of COVID-19 that is safe and free from aerosol contamination and could be an affordable and attractive choice for governments seeking to promote their emergency preparedness and better their responses to the continuing COVID-19 epidemic. In addition, this method also has great potential for early monitoring and warning of the epidemic situation at on-site-nursing points.

**IMPORTANCE** The global COVID-19 epidemic, ongoing since the initial outbreak in 2019, has caused panic and huge economic losses worldwide. Due to the continuous emergence of new variants, COVID-19 has been responsible for a higher proportion of asymptomatic patients than the previously identified SARS and MERS, which makes early diagnosis and prevention more difficult. In this manuscript, we describe a rapid, sensitive, and specific detection tool, RT-RPA-VF. This tool provides a new alternative for the detection of SARS-CoV-2 variants in a range as low as 1 to 0.77 copies/$\mu$L RNA transcripts. RT-RPA-VF has great potential to ease the pressure of medical diagnosis and the accurate identification of patients with suspected COVID-19 at point-of-care.

**KEYWORDS** RT-RPA, SARS-CoV-2, amplification, detection, viral RNA, visual

Address correspondence to Hualei Wang, wanghualei@jlu.edu.cn, Haili Zhang, zhanghaili@jlu.edu.cn, or Yuwei Gao, yuwei0901@outlook.com.

The authors declare no conflict of interest.

The causative agent of coronavirus disease 2019 (COVID-19), severe acute respiratory syndrome coronavirus 2 (SARS-CoV-2), is a positive-strand RNA virus that is classified within the genus *Betacoronavirus* (subgenus *Sarbecovirus*) of the family *Coronaviridae* (1–3). SARS-CoV-2 is more highly contagious than either SARS-CoV or Middle East respiratory syndrome coronavirus (MERS-CoV), although the viruses have a high degree of genome sequence homology (4). The World Health Organization (WHO) announced on 30 January 2020 that they had listed the COVID-19 outbreak as a public health emergency of international concern (5). Since the first case was confirmed in December 2019, the number of confirmed cases has continued to increase. As of December 2022, more than 64 million people have been infected with SARS-CoV-2 (6). Although many vaccines have been approved and marketed, due to the wide spread of the epidemic and because of the continuous evolution of the virus, the existing vaccines do not provide 100% protection for all people across the globe (7). Reliable and accurate detection tools therefore remain a powerful and necessary means to monitor confirmed and suspected COVID-19 cases and to control transmission among humans. Thus, testing strategies and multisource surveillance have been recommended as COVID-19 priorities by WHO.

WHO described the ideal method for detecting pathogens as fast, specific, sensitive, instrument free, and cost-effective (8). At present, the detection of SARS-CoV-2 focuses on nucleic acid, antigen, and serological (IgG/IgM antibody) testing, but the current mainstream method is nucleic acid detection (9). Real-time reverse transcription (RT)-PCR is still the gold standard technology for the confirmation of SARS-CoV-2 and has outstanding advantages in terms of sensitivity (10). However, traditional nucleic acid detection is generally only available to laboratories, as the technique requires precision instruments, reagents, and well-trained personnel. Hence, collected samples need to be transported to testing centers for assessment. This increases the time necessary to obtain the results and, therefore, is not conducive to the timely control of COVID-19 or the prompt treatment of infected patients (11). Although serological tests like enzyme-linked immunosorbent assays (ELISAs) have high feasibility, antibody detection has a certain lag after infection before a sample will test positive. This is because antibodies can only be detected after activation of the human immune response (12). The immunochromatographic strip used for antigen detection is not accurate enough due to the limitation of its own sensitivity, and it is also vulnerable to environmental, pH, and other factors, leading to false positives. Therefore, this method is more suitable for use in auxiliary detection.

Because of the limitations described above, these methods are not able to achieve rapid detection at the front line during COVID-19 outbreaks. In this study, we developed a simple and rapid reverse transcription-recombinase polymerase amplification with closed vertical flow visualization strip (RT-RPA-VF) assay targeting the nucleoprotein (*N*) gene of SARS-CoV-2. The assay has high specificity and sensitivity and is expected to have applications in the detection of SARS-CoV-2 in patients at grassroots sites and to be invaluable in the early diagnosis of COVID-19.

## RESULTS

**Sequence alignment of SARS-CoV-2 strains.** The RNA sequences of the *N* genes from 83 SARS-CoV-2 strains, including the ancestral strain and variant strains isolated from different countries across the world, were analyzed using WebLogo and GeneDoc. From the results of the sequence alignment (Fig. 1 and Fig. S1 in the supplemental material), combined with the targets recommended by WHO and national standards, the position 126-to-308 region of the *N* gene, which has high sequence homology between strains, was determined as the target sequence. The primers and probe were designed according to the targeting sequence. The forward primer was labeled with biotin at the 5′ end. The probe was modified to include an FITC (fluorescein isothiocyanate) label at the 5′ end, and at the 3′ end, it contained a tetrahydrofuran

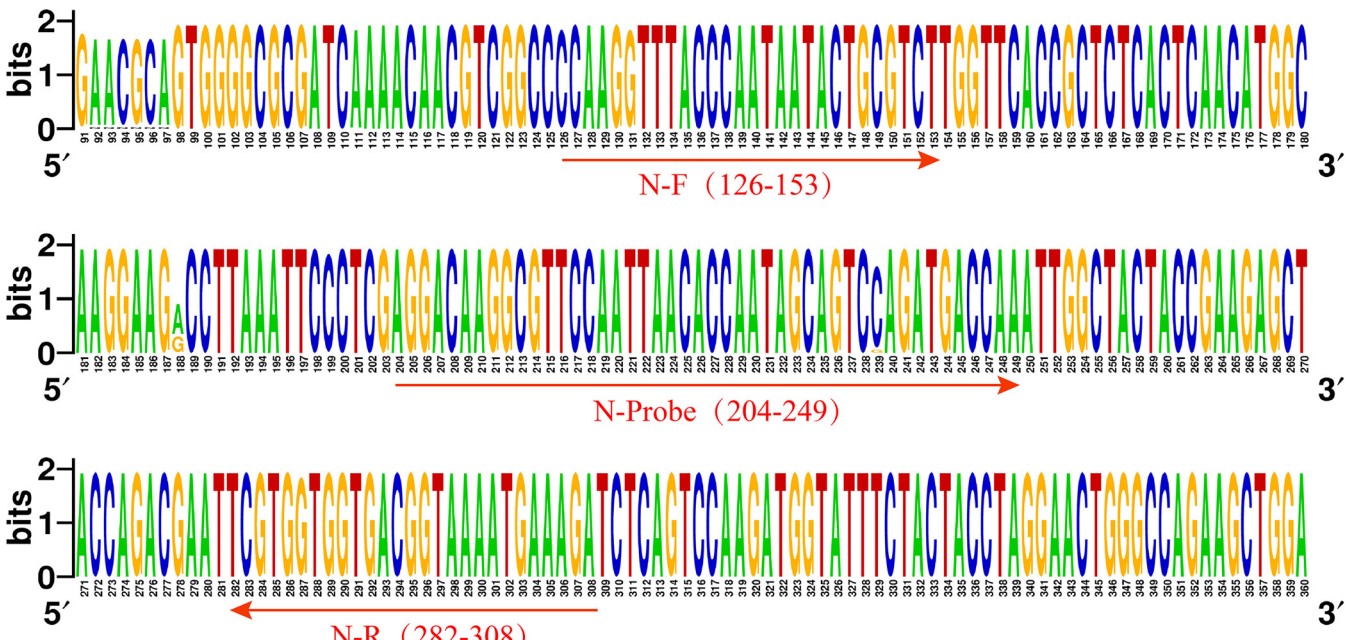

**FIG 1** Target sequence analysis of SARS-CoV-2. WebLogo presentation of the multiple alignment of *N* genes from 83 SARS-CoV-2 strains. The height of each stack corresponds to the level of nucleotide conservation at that position. The primers and probe designed in this study are labeled.

(THF) residue, which replaced a nucleotide, and a C3 spacer blocking group, making the method both sensitive and visual (Table 1).

**Optimization of the RT-RPA-VF reaction conditions.** Tenfold dilutions of the synthesized positive plasmid pUC57-N ($3.6 \times 10^{10}$ to $3.6 \times 10^{-3}$ copies/$\mu$L) were used as a template for the RT-RPA-VF assay. The reaction process was then performed at different temperatures (42°C, 40°C, 39°C, 37°C, 35°C, and 33°C) for 30 min. At a temperature of 42°C, when the plasmid template concentration was greater than or equal to 0.36 copies/$\mu$L, red bands appeared on both the control line and the test line, indicating that the test results were positive. At 37°C, 39°C, and 40°C, the results were positive only when the plasmid concentration was more than 3.6 copies/$\mu$L. Therefore, the reaction temperature was reasonable in the range of 37°C to 42°C, but 42°C was the optimal temperature (Table 2).

Similarly, SARS-CoV-2-positive plasmid standards ranging from $3.6 \times 10^{-2}$ copies/$\mu$L to $3.6 \times 10^{1}$ copies/$\mu$L were used as templates for amplification for 10 min, 15 min, 20 min, 25 min, and 30 min. The results showed that 0.36 copies/$\mu$L of positive plasmids could be detected following an amplification time of 20 min, indicating that the optimal reaction time of the RT-RPA-VF was 20 min (Table 3). Therefore, the RT-RPA-VF assay we established here showed the highest sensitivity at 42°C with 20 min for amplification.

**RT-RPA-VF sensitivity evaluation.** The sensitivity of the SARS-CoV-2 primers and probe was evaluated using the RNA transcripts and pUC57-N plasmids as templates. The RNA transcripts and plasmids were diluted 10-fold and amplified for 20 min, and the RT-RPA was conducted at 42°C. The amplified products were detected using a visual nucleic acid detection device. The results showed that the detection limits were 0.36 copies/$\mu$L for the plasmids (Fig. 2) and 1 copy/$\mu$L for the RNA transcripts (Fig. 3).

**TABLE 1** Primers and probe used in the RT-RPA-VF assay against SARS-CoV-2

| Primer name | Sequence (5′–3′) | Position |
|---|---|---|
| N-F | Biotin-CCAAGGTTTACCCAATAATACTGCGTCT | 126–153 |
| N-R | TCTTTCATTTTACCGTCACCACCACGA | 282–308 |
| N-Probe | FITC-AGGACAAGGCGTTCCAATTAACACCAATAG-THF-AGTCCAGATGACCAA-C3 Spacer | 204–249 |

**TABLE 2** Optimization of RT-RPA-VF reaction temperature

| Amt of positive plasmid used as template (copies/$\mu$L) | Result with indicated reaction temp (°C) | | | | | |
|---|---|---|---|---|---|---|
| | 33 | 35 | 37 | 39 | 40 | 42 |
| $3.6 \times 10^1$ | + | + | + | + | + | + |
| $3.6 \times 10^0$ | − | − | + | + | + | + |
| $3.6 \times 10^{-1}$ | − | − | − | − | − | + |

Because the assay was developed on a region of the *N* gene that is highly conserved between SARS-CoV-2 variants, RNA transcripts from SARS-CoV-2 Delta and Omicron variants were used for evaluation of the sensitivity of our RT-RPA-VF to different variants. The results showed that the RT-RPA-VF assay could detect RNA transcripts from the Delta variant at concentrations as low as 1 copy/$\mu$L (Fig. 4A) and Omicron variant RNA transcripts at concentrations as low as 0.77 copies/$\mu$L (Fig. 4B), indicating that RT-RPA-VF could be applied to the detection of SARS-CoV-2 variant strains.

**Evaluation of RT-RPA-VF specificity.** The NATtrol RP1 and RP2 multimarker controls kit (ZeptoMetrix Corporation, Franklin, MA, USA), which includes coronavirus (229E, OC43, NL63, and HKU1), influenza A/B virus, rhinovirus, adenovirus, parainfluenza virus, *Chlamydophila pneumoniae*, *Bordetella pertussis*, and others, was used to evaluate the sensitivity of this assay. Total RNA and DNA were obtained from the NATtrol RP1 and RP2 multimarker controls kit. As well as the RNA from SARS-CoV-2, those from the SARS-related CoV virus, MERS-CoV, porcine epidemic diarrhea virus (PEDV), and feline coronavirus (FCoV) were used as detection templates to evaluate the specificity of the RT-RPV-VF assay. The results suggested that the assay detected SARS-CoV-2 RNA, while RNAs from other coronaviruses and respiratory pathogens showed no reaction (Fig. 5). These results indicated that the RT-RPA-VF assay established in this study had no cross-reaction with other human coronavirus strains (MERS-CoV, 229E, OC43, NL63, and HKU1), adenovirus, influenza A/B virus, rhinovirus, adenovirus, parainfluenza virus, SARS-related CoV virus, PEDV, FCoV, *Chlamydophila pneumoniae*, or *Bordetella pertussis* or any other of the included pathogens, showing that the assay had high specificity for SARS-CoV-2.

**RT-RPA-VF assay evaluation of respiratory secretions from infected laboratory animals and RNA mixtures.** A total of 31 respiratory secretion samples were collected from golden hamsters, C57 mice, and cynomolgus monkeys infected with SARS-CoV-2 on the 3rd, 5th, and 7th days postinfection, and 40 respiratory secretion samples were collected from healthy animals. RNA was extracted from all samples. The whole process (from sample collection to RNA purification) was performed in a biosafety level 3 laboratory. The RNAs extracted as described above were then screened for the *N* gene using both the RT-RPA-VF assay and a COVID-19 coronavirus real-time PCR kit (BioPerfectus Technologies, Jiangsu, China). All 31 respiratory secretion samples from animals infected with SARS-CoV-2 were detected as positive with both the RT-RPA-VF and the RT-qPCR method. Furthermore, the 40 respiratory tract samples from healthy animals tested negative in all cases using both methods (Table 4). This indicates that our RT-RPA-VF assay is 100% consistent with the commercially available RT-qPCR kit for SARS-CoV-2 detection and is a feasible alternative for the clinical screening of suspected SARS-CoV-2 positive samples.

To simulate the clinical samples, hand-mixed RNAs consisting of SARS-CoV-2 (Wuhan-Hu-1 strain) RNA and RNA from throat swab samples from healthy humans were used as samples to evaluate the RT-RPA-VF assay. Amounts of $3.1 \times 10^7$, $3.1 \times 10^4$, $3.1 \times 10^2$,

**TABLE 3** Optimization of RT-RPA-VF reaction time

| Amt of positive plasmid used as template (copies/$\mu$L) | Result with indicated reaction time (min) | | | | |
|---|---|---|---|---|---|
| | 10 | 15 | 20 | 25 | 30 |
| $3.6 \times 10^1$ | − | + | + | + | + |
| $3.6 \times 10^0$ | − | + | + | + | + |
| $3.6 \times 10^{-1}$ | − | − | − | − | − |

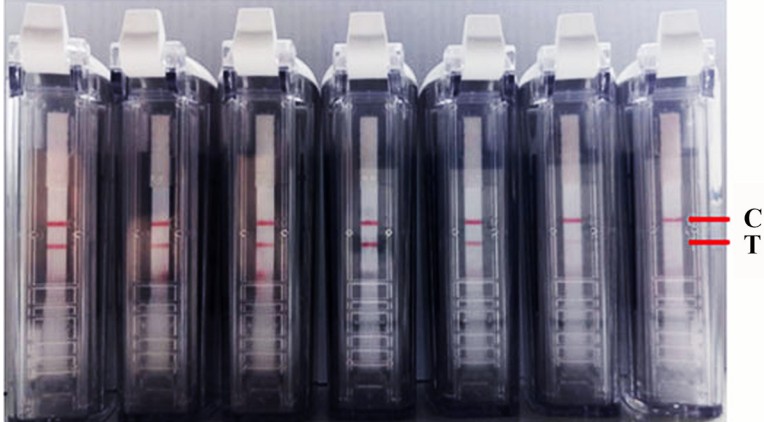

**FIG 2** Evaluation of the sensitivity of the RT-RPA-VF assay using different concentrations of recombinant plasmids. Tenfold dilutions of positive plasmid PUC57-N were used as templates for the RT-RPA-VF assay, giving a concentration range from $3.6 \times 10^{-1}$ to $3.6 \times 10^4$ copies/$\mu$L. The RT-RPA reaction was performed at 42°C for 20 min. Each dilution was independently assessed three times. T, test line; C, control line.

$3.1 \times 10^0$, and $3.1 \times 10^{-1}$ copies/$\mu$L of SARS-CoV-2 Wuhan-Hu-1 strain RNA were mixed with RNA from 11 throat swab samples in equal proportions. The 55 resultant RNA mixtures were detected by the RT-RPA-VF and RT-qPCR assays. As shown by the results in Table 4, the 55 RNA mixtures exhibited positive results by RT-RPA-VF and RT-qPCR, while RNA from the 11 healthy human throat swab samples presented negative results in all cases. The simulated clinical samples based on native RNA in the human throat secreta indirectly proved the sensitivity and specificity of the RT-RPA-VF assay.

## DISCUSSION

Over the past 20 years, disease epidemics caused by coronaviruses, such as SARS or MERS, have occurred frequently. The large-scale COVID-19 epidemic that developed following the 2019 outbreak has caused panic and huge economic losses worldwide

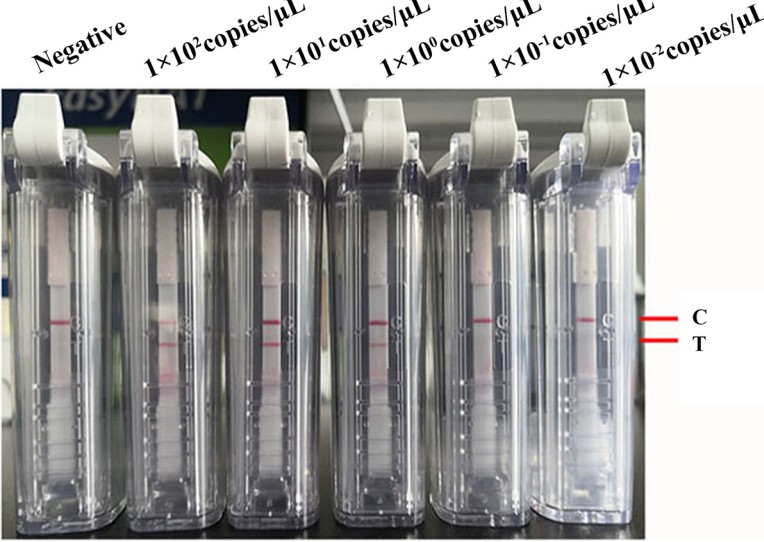

**FIG 3** Evaluation of the sensitivity of the RT-RPA-VF assay using different concentrations of RNA transcripts. Tenfold dilutions of RNA transcripts (resulting in concentrations ranging from $1 \times 10^{-2}$ to $1 \times 10^2$ copies/$\mu$L) were used to evaluate the RT-RPA-VF assay. RT-RPA reactions were performed at 42°C for 20 min. Each dilution was independently assessed three times. T, test line; C, control line.

**A**

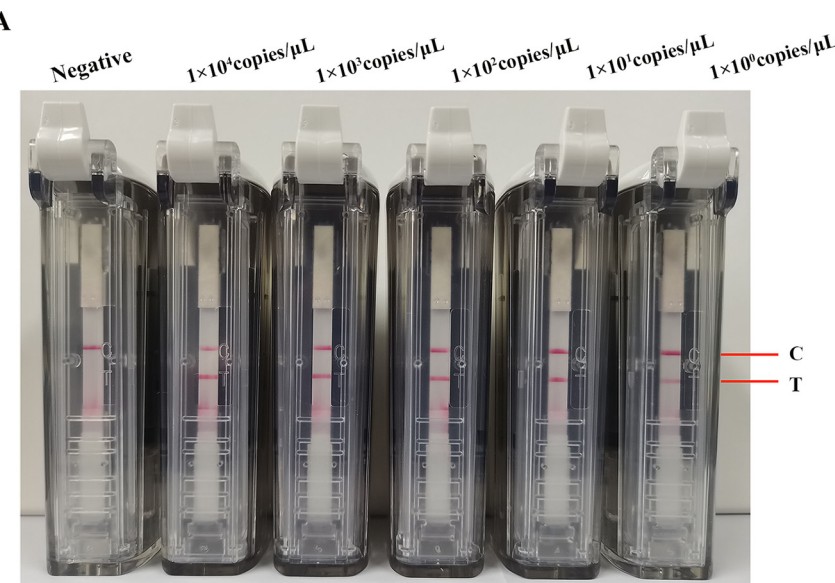

**B**

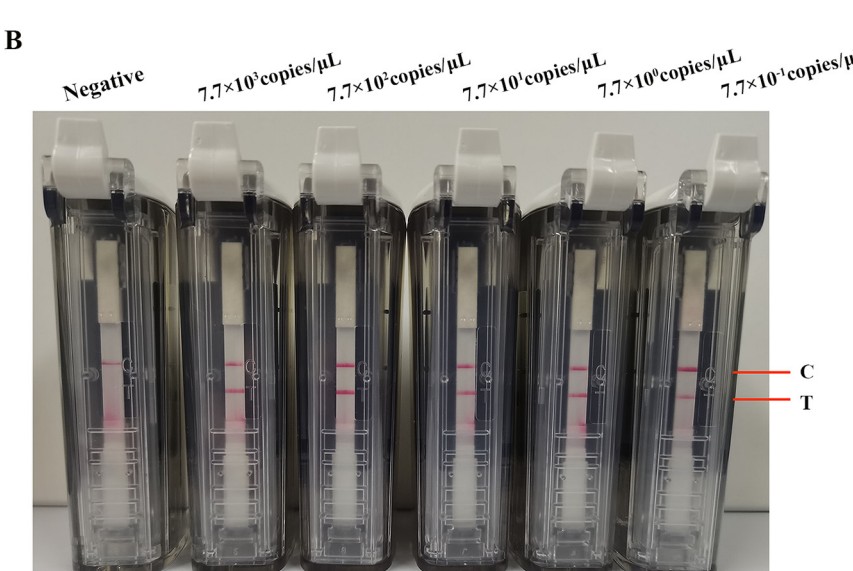

**FIG 4** Evaluation of the sensitivity of the RT-RPA-VF assay to SARS-CoV-2 variants. The detection limit of the RT-RPA-VF assay was evaluated by using different concentrations of SARS-CoV-2 Delta (4A) and Omicron (4B) variant RNA transcripts. RT-RPA reactions were performed at 42°C for 20 min. Each dilution was independently assessed three times. T, test line; C, control line.

(13). COVID-19 has been responsible for a higher proportion of asymptomatic patients; moreover, the clinical symptoms and pulmonary pathological changes in COVID-19 are not significantly different from those of other respiratory infections, meaning that COVID-19 cannot be accurately diagnosed from symptoms or medical imaging techniques alone (14). Given the difficulties in the correct identification of COVID-19 from its clinical symptoms, as well as drug shortages and incomplete vaccine protection, the development and application of new rapid, sensitive, and specific diagnostic methods are helpful in the control of the disease. Therefore, the development of a series of simple, rapid, sensitive, and specific detection tools is urgently needed, particularly for point-of-care field diagnoses.

Laboratories around the world have developed a variety of viral nucleic acid detection platforms for SARS-CoV-2 according to WHO recommendations. These platforms include RT–loop-mediated isothermal amplification (RT-LAMP), metagenomics sequencing, digital PCR, and others (15). The RT-LAMP method designed by Huang et al. is capable of detecting SARS-CoV-2 RNA transcripts at concentrations of 80 copies or more per sample

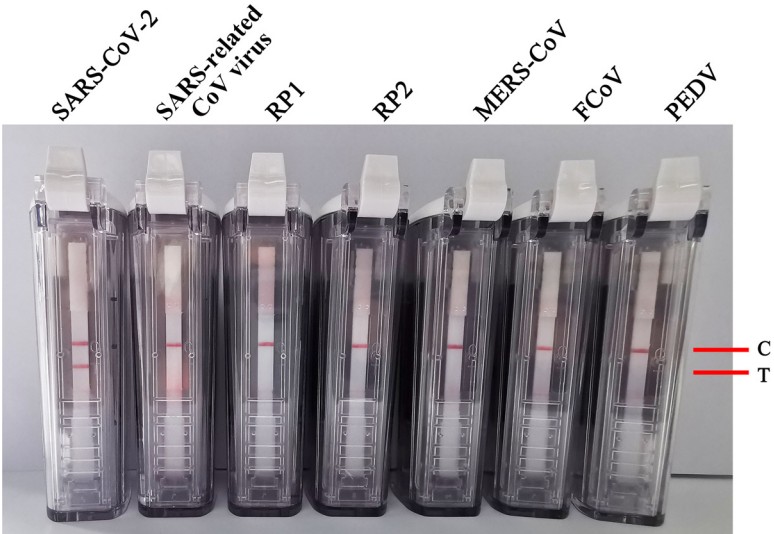

**FIG 5** The specificity of the RT-RPA-VF assay for the *N* gene from different respiratory pathogens.

at a reaction temperature of 69°C (16). Other researchers subsequently developed a CRISPR-Cas12-based approach, which, combined with RT-LAMP, can detect SARS-CoV-2 RNA directly in the samples taken from patients and without an RNA extraction step (17). However, this method requires two steps to quantify the target concentration, which may lead to the loss of the original concentration and makes cross-contamination and contamination from the environment more likely. Digital PCR is an absolute measurement method that combines limited dilution, endpoint PCR, and Poisson statistics (18). It has a lower false-negative rate than RT-PCR but is more time-consuming and more expensive than RT-PCR (19, 20). Additional novel biochemical methods are also under development. Seo et al. developed a venue-based, transistor-based biosensor that can detect SARS-CoV-2 in a nasopharyngeal swab from a COVID-19 patient in under a minute

**TABLE 4** Comparison of commercial RT-qPCR kit targeting SARS-CoV-2 *N* gene and RT-RPA-VF assay

| Specimen source | Days postinfection | Results using: | | Concordance rate (%) |
| | | RT-qPCR ($C_T$ value) | RT-RPA-VF (positive or negative) | |
|---|---|---|---|---|
| Infected laboratory animals | | | | |
| Golden hamsters (10 samples) | 3 | 28.80, 23.52, 25.81, 21.75, 36.13, 17.81, 31.13, 20.29, 30.87, 16.35 | All positive | 100 |
| C57 mice (10 samples) | 5 | 24.82, 25.28, 25.11, 25.98, 25.55, 24.90, 25.52, 25.64, 30.01, 27.31 | All positive | 100 |
| C57 mice (6 samples) | 7 | 20.07, 21.81, 13.03, 11.56, 11.20, 14.22 | All positive | 100 |
| Cynomolgus monkeys (5 samples) | 5 | 21.13, 22.82, 19.28, 21.06, 26.75 | All positive | 100 |
| | | | | |
| Healthy C57 mice, golden hamsters, and cynomolgus monkeys (40 samples) | | None | All negative | 100 |
| | | | | |
| Mixtures of RNA from 11 healthy human throat swab samples and SARS-CoV-2 RNA at a concn (copies/$\mu$L) of: | | | | |
| $10^7$ | | $C_T < 16$ | All positive | 100 |
| $10^4$ | | $22 \leqq C_T < 23$ | All positive | 100 |
| $10^2$ | | $27 \leqq C_T < 29$ | All positive | 100 |
| $10^0$ | | $33 \leqq C_T < 35$ | All positive | 100 |
| $10^{-1}$ | | $35 \leqq C_T < 39$ | All positive | 100 |
| | | | | |
| RNA from 11 healthy human throat swab samples | | None | All negative | 100 |

(21). This method can analyze patient samples directly from the buffer tubes containing cotton swabs without any sample preparation steps, and its sensitivity is about 2 to 4 times that of RT-PCR. However, the stability and application feasibility of these emerging technologies still need to be evaluated. Another study reported an ultrasensitive electrochemical biosensor based on the isothermal rolling circle amplification (RCA) method for the rapid detection of SARS-CoV-2 (22). This method involves the hybridization of an RCA amplicon and probe; however, the whole process takes about 2 h, which is too time-consuming to be used as a rapid testing method in the field.

The RPA isothermal amplification used in this study is a technique to achieve nucleic acid index amplification under constant temperature conditions with the participation of multiple proteins. The whole RPA experiment is simple and can be performed at a constant temperature, and it requires little instrumentation compared to the traditional amplification methods. With high sensitivity and specificity, this technique is suitable for rapid diagnosis at the primary level or in the field (23). Commonly used terminal interpretation methods for RPA amplification results include nucleic acid electrophoresis, turbidity indicators, and fluorescent dyes (24). In all of these methods, it is necessary to open the cover of the reaction tubes after amplification, which greatly increases the risk of environmental contamination of the amplified products. The disposable closed colloidal-gold immunochromatographic strip method used in this study removes the necessity of opening the reaction tubes, reducing contamination from the environment, and the results can be simply and rapidly observed by eye, without the use of fluorescence signal capture equipment. Moreover, this method can be performed at low reaction temperatures in a short time, making it relatively cost-effective and conducive to local rapid inspection and home self-testing.

Transmission of COVID-19 has been reported among asymptomatic people or those with only mild symptoms. The viral loads in pharyngeal swabs and sputum samples from patients reach their peaks at 5 to 6 days after onset, ranging from $10^4$ to $10^7$ copies/mL. The RT-RPA-VF assay has a minimum SARS-CoV-2 RNA transcript detection limit as low as 1 copy/$\mu$L, and it can detect RNA from the Delta and Omicron variants at 1 copy/$\mu$L and 0.77 copies/$\mu$L, respectively. Therefore, this method can be used for the early diagnosis of asymptomatic cases or those with mild symptoms to achieve early control of the transmission of COVID-19.

As the number of infected people has risen with the continuation of the epidemic, SARS-CoV-2 has continued to evolve and mutate, making the control and prevention of COVID-19 increasingly difficult (24, 25). Comparison and analysis of the *N* gene sequences of the new variant strains (b.1.1.7 strain, P.1 strain, b.1.351 strain, b.1.617.2 strain, b.1.1.529 strain, and XE strain of ba.1 and ba.2 strain of Omicron recently found in the United Kingdom), as well as the ancestral strain, revealed that the sequences targeted by the primers and probe in our method were highly conserved between different strains, indicating that the RT-RPA-VF assay has broad-spectrum applicability and is reliable. It is important to evaluate the reliability and applicability of the method using clinical samples from SARS-CoV-2-infected humans, but they were unavailable to us due to limited resources. We settled for an RNA mixture consisting of RNA from healthy human throat swab samples and SARS-CoV-2 RNA to evaluate the reliability of the RT-RPA-VF. Certainly, we expect other researchers to further validate the usefulness of the RT-RPA-VF assay in human clinical samples.

In summary, in this study we established a nucleic acid visualization assay for the SARS-CoV-2 *N* gene based on an RT-RPA-VF assay. The method does not require complex or specific experimental equipment and can be performed by simply passing the sample through a thermostatic metal bath or vacuum cup. Moreover, this assay has high specificity for SARS-CoV-2 and does not cross-react with other coronaviruses or respiratory pathogens, and we expect it to provide a strong guarantee for large-scale surveillance of the continuing SARS-CoV-2 virus epidemic.

## MATERIALS AND METHODS

**RNA transcripts and plasmids.** RNA transcripts of the partial *N* gene of SARS-CoV-2 Wuhan-Hu-1 strain were synthesized by the Sangon Biotech Co., Ltd. (Shanghai, China). A recombinant plasmid,

pUC57-N, containing the full length of the SARS-CoV-2 (Wuhan-Hu-1 strain) *N* gene was obtained from the Sangon Biotech Co., Ltd. (Shanghai, China). A bat SARS-related CoV coronavirus (SARS-related CoV virus, JTMC15 strain) was identified from bat intestinal tissue, and viral RNA of this strain was purified and then stored at −80°C. The total RNAs from MERS-CoV, HKU4, PEDV, and FCoV were taken from stocks stored in our laboratory.

**Throat swab samples and SARS-CoV-2 RNA.** Ten respiratory secretion samples were collected from infected golden hamsters on the third day postinfection. Ten and six respiratory secretion samples were collected from infected C57 mice on the fifth and seventh days, respectively. Five respiratory secretion samples were taken from infected cynomolgus monkeys on the fifth day. A total of 40 respiratory secretion samples were taken from healthy golden hamsters, C57 mice, and cynomolgus monkeys. These samples were taken from control groups at the times indicated above. The culture from VeroE6 cells infected with SARS-CoV-2 (Wuhan-Hu-1 strain) was collected and used for extracting RNA. The RNAs of all samples or of cells infected with SARS-CoV-2 were purified and processed in a biosafety level 3 laboratory of the Changchun Veterinary Research Institute. All animal experiments were approved by the Animal Care and Use Committee of the Changchun Veterinary Research Institute and were carried out by certified staff.

**Primer and probe design.** The SARS-CoV-2 *N* gene sequences from a total of 83 different strains, including the ancestral strains (18 strains), Alpha variant (5 strains), Beta variant (5 strains), Delta variant (30 strains), Gamma variant (5 strains), and Omicron variant (20 strains), were obtained from GenBank and GISRS. The sequences were aligned using Clustal X software and analyzed using the Genedoc and WebLogo website (http://weblogo.threeplusone.com). A highly conserved sequence fragment was selected as the target for designing the primers and probe (Fig. 1). The primers and probe were synthesized by the Sangon Biotech Co., Ltd. (Shanghai, China).

**The RT-RPA-VF reaction.** RT-RPA-VF reactions were conducted using a TwistAmp nfo kit (TwistDx, United Kingdom) according to the manufacturer's instructions. Briefly, reactions were run in 50-$\mu$L volumes containing 29.5 $\mu$L of rehydration buffer, 7.7 $\mu$L of diethyl pyrocarbonate (DEPC) water, 5 $\mu$L of template, 2.1 $\mu$L of both the forward and reverse primers (each at 10 $\mu$M), 0.6 $\mu$L of 10 $\mu$M probe, and 0.5 $\mu$L of 200 U murine leukemia virus (MLV) reverse transcriptase. A pipette was used to fully mix the above-described reaction mixture together with freeze-dried powder containing the recombinant enzymes, SSB and strand-displacing DNA polymerase needed for RPA amplification. Next, 2.5 $\mu$L of 280 mM magnesium acetate (MgAc) was added to the tubes. After mixing and centrifugation, the tubes were positioned in a small metal bath for amplification at a temperature range of 37°C to 42°C for 20 min. The results were read through a closed vertical flow visualization strip and became visible within 5 min.

**Real-time RT-PCR assay.** Seventy-one RNA samples from respiratory secretions from infected or healthy golden hamsters, C57 mice, and cynomolgus monkeys were assessed for the presence of SARS-CoV-2 RNA using a COVID-19 coronavirus real-time PCR kit (BioPerfectus Technologies, Jiangsu, China). All of the RT-qPCRs were repeated three times.

**Quantification of SARS-CoV-2 RNA in infected cells.** With the knowledge and permission of the subjects, 11 throat swab samples from healthy humans were collected by health care workers and soaked in phosphate-buffered saline (PBS). The supernatants were carefully transferred to sterile tubes for RNA extraction after washing by PBS, and the extracted RNA was stored at −80°C. SARS-CoV-2 (Wuhan-Hu-1 strain) RNA of infected VeroE6 cells was measured by absolute quantification RT-qPCR using the COVID-19 coronavirus real-time PCR kit. Different concentrations of *N* gene RNA transcripts were used as standards for calibrating Wuhan-Hu-1 strain RNA. All reactions were repeated two times.

**Data availability.** Further inquiries can be directed to the corresponding authors.

## SUPPLEMENTAL MATERIAL

Supplemental material is available online only.
**SUPPLEMENTAL FILE 1**, PDF file, 0.9 MB.

## ACKNOWLEDGMENTS

This study was funded by the National Key Research and Development Program of China (grant no. 2021YFF0703600), the Department of Science and Technology of Jilin Province (grant no. 20200901028SF), and the Education Department of Jilin Province (grant no. JJKH20211134KJ).

We declare no conflicts of interest associated with this study.

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
