## [Reviewer comments · Microbiology Spectrum]

Microbiology Spectrum

Rapid and visual detection of SARS-CoV-2 RNA based on RT-RPA-VF assay

Yumeng Song, Pei Huang, Mengtao Yu, Yuanguo Li, Hongli Jin, Jiazhang Qiu, Yuanyuan Li, Yuwei Gao, Haili Zhang, and Hualei Wang

Corresponding Author(s): Hualei Wang, Jilin University

Review Timeline:

Submission Date:	July 30, 2022
Editorial Decision:	September 3, 2022
Revision Received:	October 17, 2022
Editorial Decision:	October 20, 2022
Revision Received:	December 11, 2022
Accepted:	December 13, 2022

Editor: Biao He

Reviewer(s): The reviewers have opted to remain anonymous.

Transaction Report:

DOI: <https://doi.org/10.1128/spectrum.02966-22>

September 3, 2022

Prof. Hualei Wang
Jilin University
Key Laboratory of Zoonosis Research, Ministry of Education, College of Veterinary Medicine
Changchun
China

Re: Spectrum02966-22 (Rapid and visual detection of SARS-CoV-2 RNA based on RT-RPA-VF assay)

Dear Prof. Hualei Wang:

Thank you for submitting your manuscript to Microbiology Spectrum. Your manuscript has been read by two experts, both of the reviewers are concerned that your method should be evaluated using more clinical samples. I would like to consider a modified version of your manuscript.

Link Not Available

Sincerely,

Biao He

Journals Department
Reviewer comments:

Reviewer #1 (Comments for the Author):

Major Comments for the Author

This study by Song et al. aimed to assess performance characteristics of a so-called RT-RPA VF assay for rapid and visual detection of SARS-CoV-2 RNA. While this is an interesting paper, the study did not present impressive data to justify the utility of the assay being evaluated. The data presented here suggested the study is still preliminary, which decreases the enthusiasm of the reviewer. The reviewer raises concerns and questions as follows:

1. Fig 2: what is the definition of LOD? How many replicates did the authors detect for each dilutions? Merely 1: 10 dilutions are not enough to yield LOD.

2. For evaluation of specificity, what is SARS-CoV-like virus? The authors should run more respiratory viruses, bacteria and fungi to demonstrate no cross-reactivities. The current data presented are not enough.
3. For clinical evaluation, only 11 samples were run, which was insufficient.

Minor Comments for Author (Required)

None

Reviewer #2 (Comments for the Author):

Song et al established a visualized method for rapid detection of SARS-CoV-2 and its variants combining RT-LAMP and colloidal strip. The method is user friendly and requires less instruments and delicate laboratory settings, which makes it suitable for POCT fast detecting of SARS-CoV-2 and has positive meaning to the control and prevention of the disease.

Major comments:

1. What authors stated "Clinical samples" is misleading, it should be more specified. Actually, these are swabs from laboratory infected animals. To verify the method, it is highly recommended to use human clinical samples. Also, the quantity of the animal samples is not enough. Besides, the time point of sampling is very important, authors should use at least two time points for mice and three time points for monkeys. As during the first three days of the incubation period for human patients, the average detection rate of the currently used commercial RT-PCR kit is not ideal, which is a major concern to the early transmission. I suggest the authors compare their method to the commercial kit by using the samples taken from early time points.
2. Did authors try saliva samples, that would make the method more suitable for POCT.
3. The writing of the manuscript needs major improvement.

Minor comments:

1. Line 22, N gene, gene name should be italic.
2. Line 24, should be "has high sensitivity" or was highly sensitive...
3. ul should be uL, upper case L. Authors must carefully check their writings to avoid any typos, grammatical mistakes.
4. Line 44 "distinguish for ", remove "for"
5. Line 47 the description of taxonomy of SARS-CoV-2 is too ambiguous, and the literature should be properly cited.
6. Line 67, what does maturity mean here? Also, in this line "the typical nucleic acid pathogen method..." what is this method?
7. Line 75-77, better not to say "the detection rate of strips is low", they serve the different purposes, and strips are now widely used in the real world. Also please define what are "... and other phenomena"
8. Line 90 "...with high conservation..." is not correct expression, I guess the authors mean highly conserved, or with high homology?
9. Line 91-95, why these modifications make the method more sensitive?
10. Line 170, what does "...quantify the inaccurate detection..." mean?
11. Line 178, "nasophageal swab"? do authors mean nasopharyngeal?
12. Line 208-210, again, authors need test and compare more samples from different time points to make the claim.
13. Line 237-238, I'm wondering handling SARS-CoV-2 in BSL-3 facility shouldn't be approved by higher authorities other than the authors' institute?
14. Line 265-266, "grant no" or "Grant"? authors should unify this.

Staff Comments:

Preparing Revision Guidelines

For complete guidelines on revision requirements, please see the journal Submission and Review Process requirements at

<https://journals.asm.org/journal/Spectrum/submission-review-process>. **Submissions of a paper that does not conform to Microbiology Spectrum guidelines will delay acceptance of your manuscript. "**

Please return the manuscript within 60 days; if you cannot complete the modification within this time period, please contact me. If you do not wish to modify the manuscript and prefer to submit it to another journal, please notify me of your decision immediately so that the manuscript may be formally withdrawn from consideration by Microbiology Spectrum.

We thank the reviewers for their thorough review of our manuscript. We have addressed each of their comments below in a point-by-point manner:

Reviewer #1 (Comments for the Author):

Major Comments for the Author

This study by Song et al. aimed to assess performance characteristics of a so-called RT-RPA VF assay for rapid and visual detection of SARS-CoV-2 RNA. While this is an interesting paper, the study did not present impressive data to justify the utility of the assay being evaluated. The data presented here suggested the study is still preliminary, which decreases the enthusiasm of the reviewer. The reviewer raises concerns and questions as follows:

1. Fig 2: what is the definition of LOD? How many replicates did the authors detect for each dilutions? Merely 1: 10 dilutions are not enough to yield LOD.

Response: We thank the reviewer for the suggestions. This is a very good suggestion for the improvement of our manuscript. LOD is defined as the limit of detection and is used to indicate the lowest concentration of the target that can be detected by the assay. To confirm the LOD of the RT-RPA-VF method, positive plasmid containing SARS-CoV-2 *N* gene was serially diluted 10-fold from 3.6×10^4 copies/ μ L to 3.6×10^{-1} copies/ μ L and subsequently used as templates. Each dilution was independently detected three times. Revised sentence was displayed in lines 379-382: “Ten-fold dilutions of positive plasmid PUC57-N were used as templates of the RT-RPA-VF assay, giving a concentration range from 3.6×10^{-1} to 3.6×10^4 copies/ μ L. The RT-RPA reaction was performed at 42 °C for 20 minutes. Each dilution was independently assessed three times”.

2. For evaluation of specificity, what is SARS-CoV-like virus? The authors should run more respiratory viruses, bacteria and fungi to demonstrate no cross-reactivities. The current data presented are not enough.

Response: We thank the reviewer for the suggestions. We are sorry for the unclear description of SARS-CoV-like virus. SARS-CoV-like virus is animal coronavirus that is highly homologous to SARS-CoV (1-3). In this study, this bat SARS-CoV-like virus (JTMC15 strain) was used for specificity evaluation of this method. We have added the information in lines 242-244: “A bat SARS-CoV-like coronavirus (SARS-CoV-like virus, JTMC15 strain) was identified from bat intestinal tissue, and viral RNA of this strain was purified and then stored at -80 °C”. With regard to the specificity evaluation section, we verified the cross-reactivity of multiple respiratory viruses and bacteria with this RT-RPA-VF for SARS-CoV-2, but we neglected to go into a clear description of this situation. A detailed data was added in lines 130-144: “The NATtrol RP1 and RP2 Multimarker Controls kit (ZeptoMetrix Corporation, Franklin, MA, United States), which includes coronavirus (229E, OC43, NL63, HKU1), influenza A/B virus, rhinovirus, adenovirus, parainfluenza virus, *C. pneumoniae* and *Bordetella pertussis* and others, was used to evaluate the sensitivity of this assay. Total RNA and DNA were obtained from the NATtrol RP1 and RP2

Multimarker Controls kit. As well as the RNA from SARS-CoV-2, that from the SARS-CoV-like virus, MERS-CoV, porcine epidemic diarrhea virus (PEDV) and feline coronavirus (FCoV) were used as detection templates to evaluate the specificity of the RT-RPV-VF assay. The results suggested that the assay detects SARS-CoV-2 RNA, while RNA from other coronaviruses and respiratory pathogens showed no reaction (Fig. 5). The above results indicated that the RT-RPA-VF assay established in this study had no cross-reaction with other human coronavirus (MERS-CoV, 229E, OC43, NL63, HKU1), adenovirus, influenza A/B virus, rhinovirus, adenovirus, parainfluenza virus, SARS-CoV-like virus, PEDV, FCoV, *C.pneumoniae* and *Bordetella pertussis* or any other of the included pathogens, showing that the assay has high specificity to SARS-CoV-2”

3. For clinical evaluation, only 11 samples were run, which was insufficient.

Response: We thank the reviewer for the suggestions. We have increased the sample size for evaluating RT-RPA-VF assay. Ten respiratory secretions of infected golden hamster were collected on the 3rd day. Ten and six respiratory secretions of infected C57 mice were gained on the 5th, 7th day, respectively. Five respiratory secretions of infected cynomolgus monkeys were collected on the 5th day. Forty negative samples and thirty-one positive samples with different virus load were detected by RT-RPA-VF and RT-qPCR methods. The results showed that our method was consistent with the results of RT-qPCR. The data were shown in lines 147-156: “A total of thirty-one respiratory secretion samples were collected from infected golden hamsters, C57 mice and cynomolgus monkeys with SARS-CoV-2 on the 3rd, 5th and 7th day, and forty respiratory secretion samples were collected from healthy animals. RNA was extracted from all samples. The whole process (from sample collection to RNA purification) was performed in a Biosafety Level 3 laboratory. The RNA extracted as described above was then screened using both the RT-RPA-VF assay and a commercially available RT-qPCR kit for the *N* gene. All thirty-one respiratory secretion samples from animals infected with SARS-CoV-2 detected positive both the RT-RPA-VF and RT-qPCR methods. Furthermore, the forty respiratory tract samples from healthy animals tested negative results in all cases using both methods (Table 4)”.

Reviewer #2 (Comments for the Author):

Song et al established a visualized method for rapid detection of SARS-CoV-2 and its variants combining RT-LAMP and colloidal strip. The method is user friendly and requires less instruments and delicate laboratory settings, which makes it suitable for POCT fast detecting of SARS-CoV-2 and has positive meaning to the control and prevention of the disease.

Major comments:

1. What authors stated "Clinical samples" is misleading, it should be more specified. Actually, these are swabs from laboratory infected animals. To verify the method, it is highly recommended to use human clinical samples. Also, the quantity of the animal samples is not enough. Besides, the time point of sampling is very important, authors

should use at least two time points for mice and three time points for monkeys. As during the first three days of the incubation period for human patients, the average detection rate of the currently used commercial RT-PCR kit is not ideal, which is a major concern to the early transmission. I suggest the authors compare their method to the commercial kit by using the samples taken from early time points.

Response: We thank the reviewer for the suggestions. This is a very good suggestion for the improvement of our manuscript. We agree with reviewer, “Clinical samples” is inappropriate since we only used laboratory animal samples. Then we replaced the original subtitle “RT-RPA-VF Clinical Samples Evaluation” to “RT-RPA-VF Assay Evaluation of Respiratory Secretions from Infected Laboratory Animals”, **line 145**. To confirm the applicability of the method to the early stages of infection, respiratory samples were collected from infected laboratory animals at 3rd, 5th and 7th day to evaluate the consistency of the RT-RPA-VF with commercial RT-qPCR. The relevant information was supplemented in lines **147-156**: “A total of thirty-one respiratory secretion samples were collected from infected golden hamsters, C57 mice and cynomolgus monkeys with SARS-CoV-2 on the 3rd, 5th and 7th day, and forty respiratory secretion samples were collected from healthy animals. RNA was extracted from all samples. The whole process (from sample collection to RNA purification) was performed in a Biosafety Level 3 laboratory. The RNA extracted as described above was then screened using both the RT-RPA-VF assay and a commercially available RT-qPCR kit for the *N* gene. All thirty-one respiratory secretion samples from animals infected with SARS-CoV-2 detected positive both the RT-RPA-VF and RT-qPCR methods. Furthermore, the forty respiratory tract samples from healthy animals tested negative results in all cases using both methods (Table 4)”.

2. Did authors try saliva samples, human clinical samples is better that would make the method more suitable for POCT.

Response: We thank the reviewer for the suggestions. We agree with the reviewer’s view that saliva sample is easy obtained and better suitable for POCT. Considering the viral load of the sample, respiratory specimens were recommended as optimal specimen in the diagnostic testing for SARS-CoV-2 by WHO. And some studies have found that individual nasopharyngeal and oropharyngeal swabs yield a more reliable result than oral fluid specimens, especially oral fluid specimens (e.g. induced saliva), reported detection rates compared with in the upper respiratory tract specimens from the same patient vary widely (4,5). Therefore, we chose respiratory secretions to evaluate this method.

3. The writing of the manuscript needs major improvement.

Response: We thank the reviewer for the suggestions. This is a very good suggestion for the improvement of our manuscript. We have checked and modified the grammar of the manuscript again. All change was highlight in marked up manuscript

Minor comments:

1. Line 22, N gene, gene name should be italic.

Response: We thank the reviewer for the suggestions. We have corrected the format of gene names in the revised manuscript.

2. Line 24, should be "has high sensitivity" or was highly sensitive...

Response: We thank the reviewer for the suggestions. We have modified this sentence in line 25: "This method was highly sensitive comparable to RT-qPCR".

3. ul should be uL, upper case L. Authors must carefully check their writings to avoid any typos, grammatical mistakes.

Response: We apologize for the mistakes, and we have corrected this word in the revised manuscript.

4. Line 44 "distinguish for ", remove "for"

Response: We thank the reviewer for the suggestions. We have changed the content of this part to make it more clearly. The revisions are shown in lines 45-47: "RT-RPA-VF has great potential to ease the pressure on medical diagnosis and the accurate identification of patients with suspected COVID-19 at point-of-care".

5. Line 47 the description of taxonomy of SARS-CoV-2 is too ambiguous, and the literature should be properly cited.

Response: Thank you for your suggestion. We have revised the classification of SARS-CoV-2 in more detail as follows: "The causative agent of COVID-19, SARS-CoV-2, is a positive-strand RNA virus that is classified within the genus *Betacoronavirus* (subgenus *Sarbecovirus*) of the family *Coronaviridae* (1-3)", lines 49-51.

6. Line 67, what does maturity mean here? Also, in this line "the typical nucleic acid pathogen method..." what is this method?

Response: We thank the reviewer for the suggestions. We are sorry for any scientific confusion due to the ambiguity of our words. The "maturity" here means that the method of Real time RT-PCR is complete and mature. To avoid confusion for readers, we deleted the word "maturity". And we have also changed the sentence "the typical nuclear acid method..." to "traditional nucleic acid detection is generally only available to laboratories...", line 69.

7. Line 75-77, better not to say "the detection rate of strips is low", they serve the different purposes, and strips are now widely used in the real world. Also please define what are "... and other phenomena"

Response: Thank you for your suggestion. We have modified this part as: "The immunochromatographic strip used for antigen detection is not accurate enough due to the limitation of its own sensitivity, and is also vulnerable to environmental, pH and other factors, leading to "false positives". Therefore, this method is more suitable for use in auxiliary detection", lines 77-80.

8. Line 90 "...with high conservation..." is not correct expression, I guess the authors mean highly conserved, or with high homology?

Response: We apologize for the mistakes, and we have corrected the phrase "with high conservation" to "has high sequence homology", line 93.

9. Line 91-95, why these modifications make the method more sensitive?

Response: Thanks for your question. The sense primer was labeled with biotin at the 5' end, the probe was modified with FITC at the 5' end. The amplicons are indirectly labeled with FITC and biotin due to the extension of the probe and reverse amplification primer. As liquid flows on the immunochromatographic strip, amplicons labeled with biotin can bind to gold particles to form a complex. When the complexes pass over the test line, they will be captured by an anti-FITC antibody fixed on the test line, and aggregated gold particles are presented as a red band. After the antigen-antibody and receptor-ligand binding, the amplicons through gold particles to achieve visual and signal amplification, so as to improve the sensitivity of the method. Furthermore, some researchers have also demonstrated that gold immunochromatographic strips amplifies amplicons 10-fold more effectively than agarose gel electrophoresis (6).

10. Line 170, what does "...quantify the inaccurate detection..." mean?

Response: Thanks for your question. We have revised this sentence make it clear, in line 181: "...this method requires two steps to quantify the target concentration...".

11. Line 178, "nasophageal swab"? do authors mean nasopharyngeal?

Response: We apologize for the wrong word, and we have corrected this word in line 189.

12. Line 208-210, again, authors need test and compare more samples from different time points to make the claim.

Response: Thanks for your suggestion. We have increased the sample numbers in analytical specificity and sensitivity part. The data were shown in lines 147-156: "A total of thirty-one respiratory secretion samples were collected from infected golden hamsters, C57 mice and cynomolgus monkeys with SARS-CoV-2 on the 3rd, 5th and 7th day, and forty respiratory secretion samples were collected from healthy animals. RNA was extracted from all samples. The whole process (from sample collection to RNA purification) was performed in a Biosafety Level 3 laboratory. The RNA extracted as described above was then screened using both the RT-RPA-VF assay and a commercially available RT-qPCR kit for the *N* gene. All thirty-one respiratory secretion samples from animals infected with SARS-CoV-2 detected positive both the RT-RPA-VF and RT-qPCR methods. Furthermore, the forty respiratory tract samples from healthy animals tested negative results in all cases using both methods (Table 4). This indicated that our RT-RPA-VF assay was 100% consistent with the commercially available RT-qPCR kit for SARS-CoV-2 detection, and is a feasible alternative for the clinical screening of suspected SARS-CoV-2 positive samples".

13. Line 237-238, I'm wondering handling SARS-CoV-2 in BSL-3 facility shouldn't be approved by higher authorities other than the authors' institute?

Response: Thanks for your suggestion. Handling of SARS-CoV-2 in a BSL-3 facility should indeed be approved by a higher authority. All samples related to SARS-CoV-2 covered in this paper were conducted in a BSL-3 laboratory of the Changchun Veterinary Research Institute, which has been authorized to do experiments related to SARS-CoV-2. We have illustrated it in lines 253-257: "These RNA of all samples with SARS-CoV-2 were purified and processed in a Biosafety Level 3 laboratory of the Changchun Veterinary Research Institute. All animal experiments were approved by the Animal Care and Use Committee of the Changchun Veterinary Research Institute, and were carried out by certified staff".

14. Line 265-266, "grant no" or "Grant"? authors should unify this.

Response: Thanks for your suggestions. The writing format has been unified in lines 287-290.

Reference

1. Xu, L., et al., Detection and characterization of diverse alpha- and betacoronaviruses from bats in China. *Virology*, 2016. 511(1): p. 69-77.
2. Saif, L.J. and K. Jung, Comparative Pathogenesis of Bovine and Porcine Respiratory Coronaviruses in the Animal Host Species and SARS-CoV-2 in Humans. *J Clin Microbiol*, 2020. 58(8).
3. Shi, Z. and Z. Hu, A review of studies on animal reservoirs of the SARS coronavirus. *Virus Res*, 2008. 133(1): p. 74-87.
4. Yang JR, Deng DT, Wu N, Yang B, Li HJ, Pan XB. 2020. Persistent viral RNA positivity during the recovery period of a patient with SARS-CoV-2 infection. *J Med Virol* 92:1681-1683.
5. Azzi L, Carcano G, Gianfagna F, Grossi P, Dalla Gasperina D, Genoni A, Fasano M, Sessa F, Tettamanti L, Carinci F, Maurino V, Rossi A, Tagliabue A, Baj A. 2020. Saliva is a reliable tool to detect SARS-CoV-2. *Journal of Infection* 81:E45-E50.
6. Hu J, Huang R, Sun Y, Wei X, Wang Y, Jiang C, Geng Y, Sun X, Jing J, Gao H, Wang Z, Dong C. 2019. Sensitive and rapid visual detection of *Salmonella* Typhimurium in milk based on recombinase polymerase amplification with lateral flow dipsticks. *J Microbiol Methods* 158:25-32.

October 20, 2022

Prof. Hualei Wang
Jilin University
Key Laboratory of Zoonosis Research, Ministry of Education, College of Veterinary Medicine
Changchun
China

Re: Spectrum02966-22R1 (Rapid and visual detection of SARS-CoV-2 RNA based on RT-RPA-VF assay)

Dear Prof. Hualei Wang:

Thank you for revising the manuscript. After assessing this version, we found that there are still some issues raised by reviewers that have not been fully addressed.

1. There is no evaluation using clinical samples. Evaluation using swabs of infected laboratory animals cannot guarantee the method's performance.
2. There is no detailed comparison with other commercial kits to evaluate the specificity, sensitivity, accuracy, and efficiency of the method. It is recommended to compare it using other kits based on antigen and/or on RNA.
3. There are some flaws in the manuscript, which should be carefully rechecked. For example, "is except to" on line 84, did the authors mean "is expected to"? Line 87, The section title reads "Results and Discussion", but there is an extra Discussion section. "SARS-CoV-like virus" should be "SARS-related CoV".
4. Lines 56-57: The data is too old, please update it.

When submitting the revised version of your paper, please provide (1) point-by-point responses to the issues above as file type "Response to Comments," not in your cover letter, and (2) a PDF file that indicates the changes from the original submission (by highlighting or underlining the changes) as file type "Marked Up Manuscript - For Review Only". Please use this link to submit your revised manuscript - we strongly recommend that you submit your paper within the next 60 days or reach out to me. Detailed instructions on submitting your revised paper are below.

Link Not Available

Sincerely,

Biao He

Journals Department
Reviewer comments:

Staff Comments:

Preparing Revision Guidelines

Please return the manuscript within 60 days; if you cannot complete the modification within this time period, please contact me. If you do not wish to modify the manuscript and prefer to submit it to another journal, please notify me of your decision immediately so that the manuscript may be formally withdrawn from consideration by Microbiology Spectrum.

1. There is no evaluation using clinical samples. Evaluation using swabs of infected laboratory animals cannot guarantee the method's performance.

Response: Thank you for your valuable suggestion. It is important to evaluate reliability and applicability of the method using the clinical samples from SARS-CoV-2 infected human, but they are unavailable for us due to limited resources. To simulate the clinical samples, hand-mixed RNA consisting of SARS-CoV-2 (Wuhan-Hu-1 strain) RNA and healthy human throat swab RNA was used as samples to evaluate RT-RPA-VF assay. The 3.1×10^7 , 3.1×10^4 , 3.1×10^2 , 3.1×10^0 and 3.1×10^{-1} copies/ μ L of SARS-CoV-2 Wuhan-Hu-1 strain RNA were mixed with 11 throat swabs-RNA in equal proportion, respectively. The resultant 55 RNA mixtures and 11 throat swabs-RNA were detected by RT-RPA-VF and RT-qPCR assay. As shown in Table 4, the 55 RNA mixtures exhibited positive results by RT-RPA-VF and RT-qPCR, while 11 throat swabs-RNA presented negative results in all cases. The simulated clinical samples based on native RNA in the human throat secreta indirectly proved the sensitivity and specificity of the RT-RPA-VF assay. Relevant information can be added in lines 303-310: “With the knowledge and permission, 11 throat swabs from healthy human were collected by health care worker and soaked in PBS. Carefully transfer the supernatant to sterile tubes for swab-RNA extraction after washing by PBS, following RNA was extracted from the supernatant and stored at -80°C . SARS-CoV-2 (Wuhan-Hu-1 strain) RNA of infected-VeroE6 cells was quantified by absolute quantitative RT-qPCR using COVID-19 Coronavirus Real Time PCR Kit. Further notes, different concentrations of *N* gene RNA transcripts were used as standards for calibrating Wuhan-Hu-1 strain RNA”. And corresponding results were described in lines 161-170: “To simulate the clinical samples, hand-mixed RNA consisting of SARS-CoV-2 (Wuhan-Hu-1 strain) RNA and healthy human throat swab RNA was used as samples to evaluate RT-RPA-VF assay. The 3.1×10^7 , 3.1×10^4 , 3.1×10^2 , 3.1×10^0 and 3.1×10^{-1} copies/ μ L of SARS-CoV-2 Wuhan-Hu-1 strain RNA were mixed with 11 throat swabs-RNA in equal proportion, respectively. The resultant 55 RNA mixtures and 11 throat swabs-RNA were detected by RT-RPA-VF and RT-qPCR assay. As shown in Table 4, the 55 RNA mixtures exhibited positive results by RT-RPA-VF and RT-qPCR, while 11 throat swabs-RNA presented negative results in all cases. The simulated clinical samples based on native RNA in the human throat secreta indirectly proved the sensitivity and specificity of the RT-RPA-VF assay”.

2. There is no detailed comparison with other commercial kits to evaluate the specificity, sensitivity, accuracy, and efficiency of the method. It is recommended to compare it using other kits based on antigen and/or on RNA.

Response: We thank the reviewer for the suggestions. We are sorry for the unclear description of this commercial kits in the original manuscript. In fact, we compared the specificity, sensitivity, and accuracy of the RT-RPA-VF assay with the COVID-19 Coronavirus Real Time PCR Kit (Bioperfectus Technologies, Jiangsu, China) in the original manuscript (Table 4). The information related to the commercial kits was added in the revised manuscript on lines 151-154: “The RNA extracted as described above was then screened using both the RT-RPA-VF assay and a

COVID-19 Coronavirus Real Time PCR Kit (Bioperfectus Technologies, Jiangsu, China) for the *N* gene”.

3. There are some flaws in the manuscript, which should be carefully rechecked. For example, "is except to" on line 84, did the authors mean "is expected to"? Line 87, The section title reads "Results and Discussion", but there is an extra Discussion section. "SARS-CoV-like virus" should be "SARS-related CoV".

Response: We apologize for the mistakes, and we have also amended the flaws in the manuscript. The sentence “is except to have applications in the detection of SARS-CoV-2 in patients at grass-roots sites” was changed into “is expected to have applications in the detection of SARS-CoV-2 in patients at grass-roots sites” on **line 84**. The section title “RESULTS AND DISCUSSION” was revised into “RESULTS” on **line 87**. And we have corrected SARS-related CoV in the revised manuscript.

4. Lines 56-57: The data is too old, please update it.

Response: Thank you for pointing this out. We have updated the latest global data on COVID-19 infections on **lines 56-57**: As of December 2022, there are more than 64 million people infected with SARS-CoV-2.

December 13, 2022

Prof. Hualei Wang
Jilin University
Key Laboratory of Zoonosis Research, Ministry of Education, College of Veterinary Medicine
Changchun
China

Re: Spectrum02966-22R2 (Rapid and visual detection of SARS-CoV-2 RNA based on RT-RPA-VF assay)

Dear Prof. Hualei Wang:

Congratulations. Your manuscript has been accepted, and I am forwarding it to the ASM Journals Department for publication. You will be notified when your proofs are ready to be viewed.

Sincerely,

Biao He
Editor, Microbiology Spectrum
